# Design and Experiments of a Real-Time Bale Density Monitoring System Based on Dynamic Weighing

**DOI:** 10.3390/s23041778

**Published:** 2023-02-05

**Authors:** Jianjun Yin, Zhijian Chen, Chao Liu, Maile Zhou, Lu Liu

**Affiliations:** 1School of Agricultural Engineering, Jiangsu University, Zhenjiang 222013, China; 2Institute of Technology, Anhui Agricultural University, Hefei 230036, China

**Keywords:** bale balers, dynamic weighing, bale density, pressure detection, signal processing

## Abstract

Bale density is one of the main performance indicators to measure the quality of baler operation. In this study, a real-time baler bale density monitoring system was designed for the problem of difficult real-time measurement of bale density on round balers. Firstly, a weighing calculation model for the rolling and sliding stage of the bale was established, and the dynamic characteristics during the contact between the bale and the inclined surface were analyzed based on ADAMS dynamics simulation. Then, a real-time monitoring system for the bale density based on the contact pressure of the inclined surface, attitude angle measurement and hydraulic monitoring of the cylinder was constructed, and the accuracy of the weighing model was confirmed. The system was used to observe and analyze the changes in the pitch angle of the carrier table and the oil pressure in the rod chamber of the backpack cylinder during the operation of the round baler. Finally, the monitoring system was calibrated and the dynamic calibration equations were obtained. The results show that the maximum error between the calculated value of the original weighing model and the actual weight was 3.63%, the maximum error of the calculated value of the weighing model corrected by the calibration equations was 3.40% and the measurement accuracy could be satisfied. The results show that the system was highly accurate and met the practical needs of bale weighing in the field.

## 1. Introduction

Straw has long been an important source of pollution due to the lack of reasonable and effective use of straw, with large amounts of straw being discarded or burnt in the wild, becoming a major source of air quality degradation [1,2]. Straw in its natural state is characterized by its low density and large size, making it difficult to collect, and it takes up excessive space for storage. The current solution to this problem is to use straw harvesting machinery to compress it into bales of a certain shape and density [3,4]. The bale density is a key parameter affecting bale quality, especially silage quality [5]. Higher bale density not only increases storage capacity but also reduces the risk of open fires [6]. In this context, weighing the characteristics of the baler is important. In practice, bales are weighed using traditional weighing methods, using forklifts and scales after the bales have been dropped on the ground, requiring multiple devices and manual operations, which is time-consuming, laborious and can easily cause damage to the bales or even cause them to fall apart. Therefore, it is important to investigate real-time bale density monitoring systems to obtain information on the variation in bale weight in the field, which is important for the study of the yield distribution of forage and straw bales.

Shinners et al. [7] proposed a dynamic weighing system on the bale chute of a large square-bale baler to monitor changes in the mass of the bales, but calibration curves are required for each cross-plate position, and parametric measurements of the slope of the ground are also required. Sun et al. [8] proposed an improvement of the traditional method of data analysis using a penetrometer to assess the density of bales by combining four round bales from a fixed cavity baler and two-dimensional data under a map, and the results showed that the quality of the round bales could be better analyzed. Maguire et al. [9] carried out a design of a dynamic weighing system for square bale weight and an experimental study of the effect of driving speed on bale weight detection, and found that the driving speed of the square bale machine had no significant effect on the accuracy of bale weight detection when the machine was driven at a speed range of 6–10 km/h. Loftis et al. [10] developed a weighing system for a round bale machine equipped with a hydraulic kick plate, in which the predicted weight of a PVC bale body was tested with two sensors at three different positions of a spring-loaded ramp model. The results of reproducible experiments in the laboratory showed better predictions using a ramp gyroscope, and by correlating the uniaxial response with the weight of the bale body, the accuracy of gyroscope was further improved to an absolute error of 3.6%, but further field testing studies are required. Zhang et al. [11] designed a dynamic weighing system for square balers based on multi-sensor fusion technology, which showed that the maximum relative error of the predicted bale quality was 0.38% in static mode, and the relative error between the predicted and true bale quality was −4.40–4.30% in dynamic mode, which provides a rapid measurement means for the evaluation of the quality of baler operation. Shang et al. [12] developed a remote monitoring and control system for a round baler to improve the automation level of the baler. The test results showed that the average time per bale was 77.93 s in the manual mode and 66.31 s in the automatic net feeding mode. It is important to mention that it is essential to improve the operational efficiency of the baler, and more research can be carried out in the future, for example, related to unmanned baling. Guo et al. [13] designed a bale weight monitoring system using a 4KZ300 self-propelled square baler as a research prototype to address the lack of bale weight monitoring in existing square baler models. Field baling tests were carried out to verify the reliability of the bale monitoring system and the results showed that the minimum percentage error in monitoring quality was 1.28% and the maximum was 4.62%, both being less than 5%.

The literature review showed that the weighing characteristics of straw bales under different conditions have been studied to some extent. However, most of the research on the weighing of baler bales has been conducted on square balers and less on the weighing of round balers. In addition, considering the significant differences between dynamic weighing and traditional weighing, it is difficult to achieve real-time automatic detection of bale density with traditional weighing. Furthermore, there is no in-depth analysis of the movement state and mechanical model of the bale during dynamic weighing. Therefore, an in-depth study of the bale mass change and dynamics during the bale weighing process, combined with baling weighing tests, was conducted to analyze the sensor detection results under different conditions, which is of great significance for the control of bale density.

This research study aimed to design a dynamic weighing system for round-baler bales through pressure detection and inertia measurement to solve the problem of weighing round-baler bales. Firstly, we conducted the analysis of the key components of the bale weighing and the forces applied during the weighing process, the mechanical relationship between the load cell output and the actual weight of the bale, and the simulation of the rolling process of the bale using ADAMS software. Secondly, the calibration equations of the bale density real-time monitoring system and the dynamic bale weighing model were completed, and the hydraulic signal and attitude angle changes in the cylinder during the calibration test were analyzed in depth. Finally, validation tests of the dynamic bale weighing model were carried out on bales weighing 280.1 kg and 405.3 kg.

## 2. Materials and Methods

### 2.1. Principle and Analysis of Round Bale Weighing

#### 2.1.1. Structure of the Weighing and Testing System

The overall structure consists of a tractor and a baler, as shown in Figure 1. The baler was mounted behind the tractor through a three-point suspension and the weighing equipment was mounted at the lower end of the baler to receive signals for accurate weighing. In actual field operation, the speed of the baler was relatively constant and its forward speed could be adjusted according to the amount of straw on site. The current traction speed of mainstream balers is generally 4–6 km/h [14]. The baler selected for this research was a WDB800 round-bale baler with a forming chamber size of Ø1300 × 200 mm and a bale volume of 1.6 m^3^. The weighing carrier table was installed at the original skidding frame position, allowing the weight of the bale to be checked while the bale was rolling down.

To minimize the impact of the bending deformation on the test results caused by the impact of the bale, 80 × 50 × 5 mm unequal angle steel was selected as the material for the fixed frame of the bearing table. When the bale contacted the bearing platform, it formed a stable platform. The bale length was 1.2 m. To ensure effective contact between the bale and the data acquisition platform during dynamic weighing, the width of the platform was set to 1 m and the size of the data acquisition platform was 1000 × 600 mm. The platform returned to its equilibrium position after the bale fell to the ground. The data acquisition platform had 4 support points, so 4 load cells were used to detect the pressure signal. At the same time, the bale should be dynamically weighed during the dynamic weighing process. The structure of the load-bearing table is shown in Figure 2.

#### 2.1.2. Analysis of Dynamic Weighing Models

Based on the bale unloading process shown above, the bale was weighed during the rolling process along the inclined surface of the swinging frame, considering the elastic deformation of the bale and analyzing the changes in force and acceleration of the bale during the rolling process after the bale had touched the inclined surface. When the bale was weighed dynamically, the carrier table was not horizontal, its attitude changed and there was a deflection between the carrier coordinate system and the geographical coordinate system of the carrier table itself, as shown in Figure 3. In the figure, the *XYZ* axis is the geographic coordinate system; the *X*_1_*Y*_1_*Z*_1_ axis is the carrier coordinate system; *Ψ* is the yaw angle, rotated around the *X* axis; *θ* is the pitch angle, rotated around the *Z* axis; and *γ* is the cross roll angle, rotated around the *Y* axis.

When the bale rolls along the data acquisition table surface, the data acquisition table surface is subjected to the pressure *F_N_* from the bale, the rolling friction f when the bale rolls, and the support force *F_N1_* from the fixed frame of the table. The direction of *F_N_* is the negative direction of the *Z*_1_ axis, and the direction of the friction f is the direction of the *X*_1_ axis. The magnitude of the support force *F*_*N*1_ is equal to the *F_N_* plus the part of the data acquisition table surface’s weight in the *Z*_1_ axis. The angle between *F_N_* and the gravity of the bale at this point is the pitch angle *θ* of the table, and the pressure *F_N_* on the table surface is the component force of the gravity of the bale in the *Z*_1_ direction. Thus, the gravity of the bale during the dynamic weighing of the bale is equal to the pressure collected on the data acquisition table divided by the cosine of the pitch angle *cos θ*. The mathematical model for the dynamic weighing of the bale is:(1)$$W=\frac{F_{N}}{\cos\theta }$$

### 2.2. Dynamics Analysis of the Bale Unloading Process

To verify the accuracy of the mathematical model for the dynamic weighing of straw bales, the model was simulated in this research based on ADAMS software [15], using SolidWorks software to build a 3D model of the dynamic weighing table, and then imported into ADAMS software to generate a dynamics model [16]. To simplify the model and to ensure that the height of the center of the bearing table spindle corresponds to the actual dimensions, and to realize the movement of the model of the bearing table, a stationary base was used instead of a round baler which was coaxially connected to the dynamic weighing bearing table spindle.

The simplified dynamic weighing table model was fixed to the ground through a fixed base; fixed constraints were added between the ground and the ADAMS ground, between the ground and the fixed base, and between the mutually fixed parts, and rotational constraints were added between the table spindle and the fixed base [17]. To make the bale unloading simulation process consistent with the actual process, so that the bale fell to the ground after the bearing table under the action of the spring swinging back to the initial position, we needed to add a spring in the bearing table fixed frame. After a lot of debugging to determine the spring upper and lower connection point to establish the spring, the bearing table model was developed, as shown in Figure 4.

During the unloading process, the bale was dropped onto the carrier stand by gravity. Then, the bale and the carrier stand swung around the axis until the carrier stand was in contact with the ground, and the bale rolled along the carrier stand to the ground. Contact constraints were added between the bale and the carrier stand, the bale and the data acquisition table surface, the carrier stand and the ground, and the bale and the ground. The bale fell freely onto the carrier stand, and there was no need to add a drive in the simulation. The material parameters of the bale are defined in Table 1.

To make the simulation results more accurate, a mechanical structure was used to continuously change the inclination angle *θ* of the inclined plane to measure the rolling friction coefficient of the bale. The bale was placed on the inclined surface with an inclination angle of *θ*. When the inclination angle is small, the bale is stationary on the inclined surface, and the friction force at this time is the static friction force [18]. Varying the magnitude of *θ*, the equation of motion of the bale rolling along the inclined plane under the action of gravity, support force, and friction is:(2)$$\left\{\begin{array}{l} \sum F=Ma\\ \sum M=I\alpha \\ I=\frac{1}{2}MR^{2}\\ a=R\alpha \end{array}\right.$$
where *R*—the radius of the bale, *α*—the angular acceleration, *a*—the linear acceleration, *I*—the rotational inertia, *M*—the weight of the bale and *F*—the combined external force. Force analysis of straw bales entails:(3)$$\left\{\begin{array}{l} Mg\sin\theta -f=Ma\\ f_{R}=I(a/R) \end{array}\right.$$

Solving these yields:(4)$$f=\frac{1}{3}Mg\sin\theta ;a=\frac{2}{3}g\sin\theta$$

As the inclination angle *θ* of the inclined plane is variable, each angle *θ* corresponds to a different state of motion. From Equation (2), it follows that:(5)$$\left\{\begin{array}{l} f_{\max}=\mu_{s}Mg\cos\theta ;f_{k}=\mu_{k}Mg\cos\theta \\ \alpha =\frac{2\mu_{k}g\cos\theta }{R};a=g\left(\sin\theta -\mu_{k}\cos\theta \right) \end{array}\right.$$
where *f_max_*—maximum static friction, *μ_s_*—the coefficient of static friction, *f_k_*—the rolling friction force *f_k_* and *μ_k_*—the coefficient of kinetic friction.

During the movement of the bale, the distance between the bale rolls and slides is indicated by *X_L_*, and the position of the bale’s movement at different times is recorded.
(6)$$X_{L}=\frac{1}{2}at^{2}=\frac{1}{2}g\left(\sin\theta -\mu_{k}\cos\theta \right)t^{2}$$

The rolling distance of the bale is *X_Ω_* from the angle of rotation, and there is a difference Δ*X* between the measured value *X_L_* and the calculated value *X*.
(7)$$\Delta X=X_{L}-X_{\Omega }=\frac{1}{2}g\left(\sin\theta -3\mu_{k}\cos\theta \right)t^{2}$$

The distance the bale slides over after *Ν* turns is therefore Δ*X_Ν_*, with Δ*X_Ν_* corresponding to the different positions where the bale is located.
(8)$$\Delta X_{N}=N\pi R\left(\frac{\tan\theta }{\mu_{k}}-3\right)$$

We measured and recorded each movement on an inclined surface, with the distance between successive marks being *D*.
(9)$$D=X_{L(N+1)}-X_{LN}=2\pi R+\pi R\left(\frac{\tan\theta }{\mu_{k}}-3\right)$$

Finally, the D-tan *θ* variation relationship can be plotted to obtain the slope of the line m to determine the kinetic friction coefficient *μk*.
(10)$$\mu_{k}=\frac{\pi R}{m}$$

### 2.3. Design of a Real-Time Bale Density Monitoring System for Round Balers

#### 2.3.1. Bale Density Real-Time Monitoring System Equipment

In addition to the pressure from the bale, the sensor was also affected by self-weight, off-load and vibration during the bale weighing process. The instantaneous load on the sensor was 10% to 40% greater than the actual load, even exceeding the rated load, affecting the accuracy of the measurement [19,20]. In practical engineering applications, the following equation is used:(11)$$W_{\max}+W\leq CN\times 70\%$$
where *C*—the rated range of a single sensor; W—the self-weight of the scale; *W_max_*—the maximum value of the net weight of the object to be measured; *N*—the number of sensors; *K*—the insurance factor, taken as 0.7 to ensure that the sensor works within the rated load cell range to ensure the accuracy of the detection, taking into account the above-mentioned influencing factors to specify the upper limit of the rated load cell.

The maximum mass of the bale is 500 kg and the weight of the weighing body is 17 kg, so the range of the weighing body should be greater than 184.64 kg. There are harsh operating conditions in the field, where the bale is not yet weighed and the sensor is still affected by factors such as the vibration of the round baler body [21]. The MIK-LCK series of cantilever beam load cells from Miko Sensors were selected, with a range of 200 kg. The parameters to be collected by the bale density monitoring system included acceleration and attitude changes during the weighing process, in addition to the pressure information of the loading platform. The oil pressure sensor was used to detect the change in oil pressure in the rod cavity of the backpack cylinder during the bale-forming process. The oil pressure sensor selected in this research was the MIK-P300 pressure transmitter from Miko Sensors, Germany, and the system was selected based on the ARM series STM32 processor as the core processor. The processor system receives the data in real time through the port, as shown in Figure 5, which depicts the overall structure of the real-time monitoring system. 

#### 2.3.2. Bale Density Real Time Monitoring System Software System

After integrating the information from the three components, the bale density monitoring system for round balers was run in an interface developed in the Real View MDK5 development environment [22]. The main program of the bale density monitoring system was designed to initialize the system, detect the oil pressure, detect the load cell and the inertial measurement unit signals, and process, store and display the data. The initialization of the system includes data initialization, port initialization and display initialization, where the data initialization was the zero calibration of the dynamic bale weighing system, taking the average of 500 data points from the start of the initialization as the zero point of the current bale weighing system. If the monitoring system does not receive the full signal triggered by the travel switch when the bale is full, the load cell and inertial measurement unit do not collect data. Only when the full signal is triggered does the STM32 processor start to collect the load cell and inertial measurement unit signals. The whole unloading process of the bale can be completed within a few seconds, so we set the data collection amount to 4000, when the data acquisition time is 20 s and the sampling frequency is 200 Hz, which can fully satisfy the effective data acquisition time, as shown in Figure 6, which depicts the main flowchart of the bale density monitoring system.

The signals collected by the weighing sensor and pressure transmitter were analog signals, and the monitoring system chose DMA transmission for signal acquisition. During data acquisition, the GPIO port needed to be configured, and ADC acquisition in the monitoring system needed to achieve real-time acquisition as much as possible by setting the ADC working mode, the number of conversion channels and other parameters in the ADC structure. Therefore, the acquisition mode needed to be set to continuous conversion mode. There were five ADCs in the monitoring system that needed to be collected at the same time, so the number of conversion channels was set to 5. The communication mode between the AH100B navigation system and the STM32 microprocessor was an RS-232 serial port communication. The serial port level was TTL level and the serial port baud rate was 11520. In this paper, data sent and received in STM32 were realized through the data register USART_DR.

### 2.4. Bale Density Real-Time Monitoring System Testing and Experiments

#### 2.4.1. Backpack Cylinder Oil Pressure Detection during Round Baler Operation

The bale density of the round baler is controlled by adjusting the relief pressure of the relief valve in the backpack hydraulic system of the round baler [23]. The higher the relief pressure, the higher the bale density; therefore, it is necessary to understand the change in oil pressure in the backpack hydraulic system during the bale-forming process of the round baler. The round baler backpack hydraulic system is shown in Figure 7. The opening and closing of the round baler backpack are controlled by the two backpack cylinders on the left and right. When the backpack hatch of the round baler needs to be opened, the tractor hydraulic system supplies oil to the rodless chamber of the backpack cylinder, and the hydraulic check valve connected to the rodless chamber of the backpack cylinder opens during the oil supply process, and the oil in the rodless chamber flows back to the tractor hydraulic system. When the oil pressure in the rod chamber of the backpack cylinder reaches the relief valve pressure, the relief valve unloads, and the adjustment range of the relief valve is 7–18 MPa.

#### 2.4.2. Straw Bale Dynamic Weighing Model Calibration Experiment

To test the actual performance of the bale density real-time monitoring system, a field trial was conducted. The test was carried out at the test site of Jiangsu World Heavy Industries Co., Ltd., Nantong, China. The round bale machine Ward WDB800 was used as the test prototype. Before calibrating the straw bale dynamic weighing model, a foot was placed on the straw bale dynamic weighing bearing table to see if the system output was normal and to check if the system was operating properly. As shown in Figure 8, the rice straw was laid out in several columns, imitating the field operation. The calculated values of the straw bale dynamic weighing model were corrected by testing the measurement results.

#### 2.4.3. Straw Bale Dynamic Weighing Model Validation Experiments

To test the monitoring accuracy and stability of the monitoring system, bales of 403.5 kg and 280.1 kg were slowly fed into the bale outlet of the round baler backpack using a forklift truck and rolled off the bale dynamic weighing table in a state close to the actual discharge of the bale. The repeatability test was carried out as shown in Figure 9.

## 3. Discussion and Results

### 3.1. Analysis of Dynamics Simulation Results

The brush was glued to one end of the bale and the tip of the small brush was in contact with the inclined surface while the bale was in motion so that the bale could roll freely along the inclined surface when the cylinder was on the inclined surface, by adjusting the inclination angle *θ* of the inclined surface and measuring its value, releasing the cylinder at the top of the inclined surface and letting the cylinder roll freely on the inclined surface. The point where the brush first comes into contact with the inclined surface is the first marked point. The distance of each mark on the surface of the inclined plane from the first marked point was measured. The first distance is recorded as *X*_*L*1_ and the second distance is recorded as *X*_*L*2_. The above process was repeated and the final results of the process are shown in Figure 10. When *θ* is greater than the critical angle *θc*, the fitted D-tan *θ* relationship is a straight line with slope *m* = 337.9, and then *μ_k_* = 0.0929 from Equation (10).

With the contact constraints and parameters set in the model, the simulation results in Figure 11 showed the contact position and motion of the bale with the different parts of the loading table during the bale unloading process, simulating the motion and kinetic behavior of the bale unloading process. The main purpose of the dynamics simulation was to observe the movement of the bale during the unloading process and the relationship between the pressure from the bale and the weight of the bale on the data collection table, to verify the correctness of the mathematical dynamic bale weighing model and to provide a theoretical basis for the design of a dynamic bale weighing system for round balers.

### 3.2. Bale Density Real-Time Monitoring System Testing and Experiments (Analysis of Dynamics Simulation Results)

Figure 12a shows the pressure on the data acquisition table and the acceleration of the data acquisition table during the bale unloading simulation. The graph shows that the fluctuation in pressure on the data acquisition table surface of the carrier table corresponds to the trend in acceleration. The reason for this is that the bale falls from a certain height onto the bearing table and then rolls down the table. As a result, there are obvious fluctuations in the positive pressure on the data acquisition table and acceleration fluctuations on the table, both of which are in line with each other. At this point, the data collected on the data acquisition table are mixed with noise from the bouncing bales, which can be removed through digital filtering. Figure 12b shows the angle of rotation between the carrier coordinate system of the bearing table itself and the geodetic coordinate system in the ADAMS software; the red curve is the angular velocity change curve around the axis of rotation, and the blue curve is the angular velocity integration after the angle change curve. As can be seen from the graph, the angle of the carrier table in the initial state is 10°, and the maximum pitch angle θ of the carrier table rotating around the geodetic coordinate X-axis when the bale rolls to the data collection table is 25.5°, so the pitch angle change value is 15.5°.

### 3.3. Analysis of Test Results of a Real-Time Bale Density Monitoring System

#### 3.3.1. Analysis of Hydraulic Cylinder Test Results

The change in oil pressure in the rod chamber of the backpack cylinder during the operation of the round baler is shown in Figure 13. When the bale is full and the backpack door is squeezed, the oil pressure in the rod chamber increases to 12 MPa, the bale full travel switch is triggered, the relief valve connected to the rod chamber unloads and the oil pressure in the rod chamber plummets. When the bale door is opened, the oil pressure in the rod chamber drops to 0 MPa and the bale is released.

After the pressure of the relief valve is set, the density of the straw bale being formed still fluctuates. The reason is that the feeding of the round straw bale machine is not uniform, which leads to uneven distribution of layered straw in the process of forming straw bales, and various bale types, such as those which are dense in the middle and dense on both sides, or dense on one side, may be formed along the bale bus. When the denser part of the bale acts on the steel roller to make the oil pressure of the backpack cylinder reach the pressure set by the relief valve, the relief valve unloads, the backpack cylinder elongates and triggers the travel switch, the controller receives the travel switch signal to make the buzzer alarm and the net-wrapping device starts to wrap the net around the bale. After the net-wrapping is finished, the buzzer stops alarming. The driver operates the backpack oil circuit reversing valve to make the piston rod of the backpack cylinder extend, and the rear bin opens to drop the bale. Therefore, the uneven feeding makes the distribution of the grass core uneven, and the density of the formed bale varies.

#### 3.3.2. Analysis of the Calibration Results of the Straw Bale Dynamic Weighing System

The load-bearing table was not in a horizontal static state when the bale was weighed dynamically but rather was first rotated around the tail shaft of the baler to the end contact with the ground when the pressure of the bale was applied. In addition, the body of the baler is in a constant vibration state during the operation of the baler. Thus, there are many disturbing factors in the actual weighing signal. In this paper, the signal was filtered using a FIR low-pass filter, and the filtering results were shown in Figure 14. The average of the 20 data points before and after the maximum value of the pressure data on the data acquisition table is taken as the pressure acquisition value of the data acquisition table of 2680 N. The pressure acquisition value of the data acquisition table in the simulation is divided by cos θ to be the gravity value of the bale of 2957 N. This is converted to a weight of 301.5 kg, which is consistent with the actual weight of the bale of 300 kg, proving that Equation (1) is correct and can be used as a mathematical model for a dynamic bale weighing system.

The test site is shown in Figure 15a. The tractor was operated by the driver to pull the prototype equipped with the round bale machine bale density monitoring system to work along the straw column at the same speed. Different straw feeding times were controlled to obtain ten different weights of bales, and the calibration test of the dynamic weighing model in the round bale machine bale density monitoring system was carried out.
(12)Y=0.9886X+3.7136
where *X* is the calculated value of the dynamic weighing model; *Y* is the actual weight of the bale.

The data collected by the attitude system are shown in Figure 16. During the unloading of the bales, the angle of the cross-roll and yaw of the data collection table varied little and the angle of pitch varied the most. During the pendulum rotation of the loading platform, the roll angle fluctuates from −177.85° to −177.55°, with an angle change of only 0.3°; the yaw angle fluctuates from −4.15° to −5.63°, with an angle change of only 1.48°; and the pitch angle fluctuates from 0.01° to −14.68°, with an angle change of 14.67°, which is consistent with the results of the ADAMS simulation.

The red curve shown in Figure 17 represents the pressure signal collected on the data acquisition table after being filtered by FIR low-pass filter, which matches the trend of the pressure data curve in the ADAMS dynamics simulation of the bale unloading process.

#### 3.3.3. Repeatability Validation Tests of a Dynamic Weighing Model for Straw Bales

As shown in Figure 18, the calculated values are the calculated bale weights calculated by the dynamic weighing model, and the measured values are the actual measured values calculated by the dynamic weighing model and then corrected by the calibration equation. In the repeatability test, for a bale with an actual weight of 280.1 kg, the maximum error between the dynamic weighing model and the actual weight was 3.63%, with an average error of 2.28%, and the maximum error between the actual measured value after correction of the calibration equation was 3.40%, with an average error of 2.17%. The maximum error between the calculated value and the actual weight of a 403.5 kg bale was 2.63%, with an average error of 1.09%, and the maximum error of the actual measured value corrected by the calibration equation was 2.38%, with an average error of 1.13%. The test results showed that the maximum error of the bale density monitoring system is 3.63% and the detection accuracy of the system is less than 5%, which meets the accuracy requirements.

## 4. Conclusions

The aim of this study was to design a dynamic weighing system for round bale machines based on pressure and attitude detection, as well as to determine the accuracy and robustness of a real-time bale density monitoring system, and to establish that the system can meet the practical needs of bale weighing in the field. 

(1)A dynamic bale weighing method based on pressure detection and attitude angle was proposed, and the structural design of the dynamic bale weighing table was completed. The dynamics of the bale unloading process were simulated based on ADAMS, and the motion of the bale and the load table during the unloading process and the change in the pressure, acceleration and attitude angle curve of the load table were analyzed. The dynamic bale weighing system based on the real-time down pressure and pitch angle of the weighing table was built based on multi-sensor fusion technology, and the dynamic weighing of the bale was realized.(2)We observed and analyzed the changes in the attitude angle of the loading table and the oil pressure in the rod chamber of the cylinder, and determined the reasons for the changes in the density of the formed bales. We completed the calibration test of the dynamic weighing system and obtained the calibration equations for the dynamic weighing system, and carried out repeatability verification tests on bales with actual weights of 280.1 kg and 405.3 kg. The maximum error between calculated and actual weight in the two sets of repeatability tests was 3.63%, and the maximum error after correction of the calibration equation was 3.40%, which is in accordance with the measurement accuracy requirements.(3)This research provides a solution for real-time monitoring of bale density, laying the foundation for the development of a baling agricultural robot. Further aspects of this research will continue, such as the development of dynamic weighing piezoelectric sensors on roads.

## Figures and Tables

**Figure 1 sensors-23-01778-f001:**
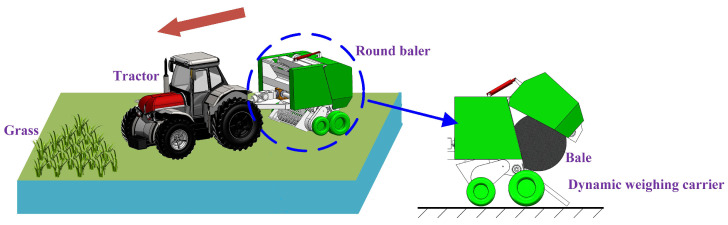
Diagram of bale collection and unloading.

**Figure 2 sensors-23-01778-f002:**
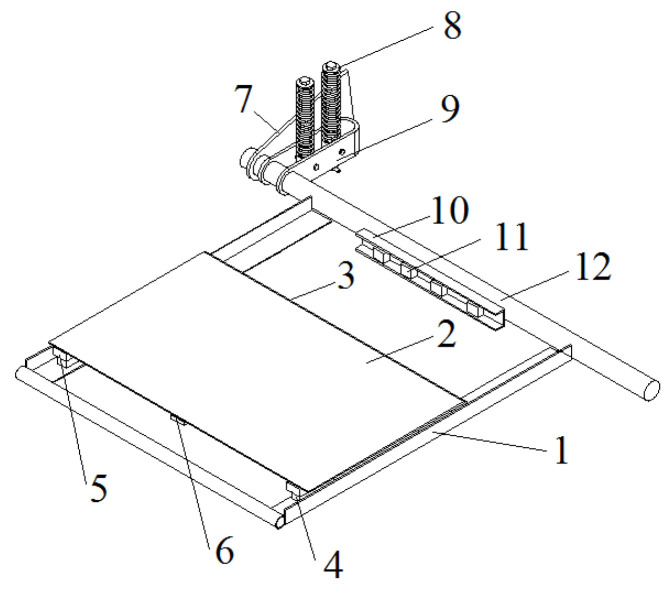
Structure of dynamic weighing carrier. (1) fixing frame; (2) table; (3) buffer arc; (4) load cell; (5) load-bearing block; (6) inertia-measuring unit; (7) connection plate; (8) double-spring mechanism; (9) u-shaped connection; (10) tail shaft; (11) signal amplifier; (12) mounting slot.

**Figure 3 sensors-23-01778-f003:**
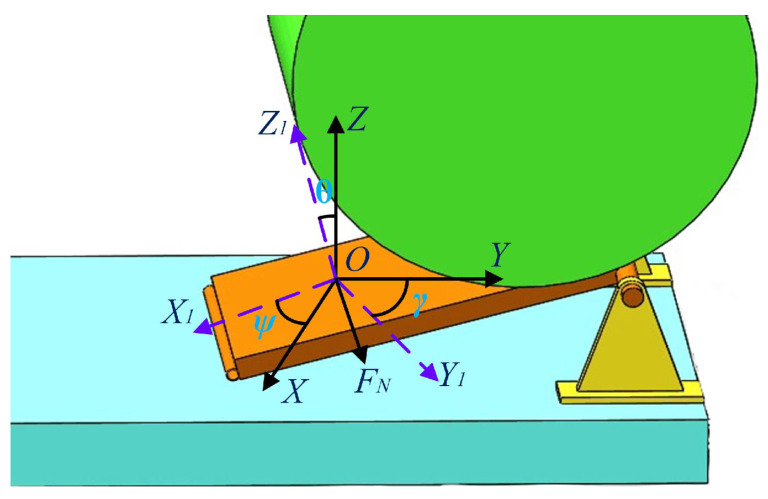
Diagram of the force analysis of the load-bearing table during dynamic weighing.

**Figure 4 sensors-23-01778-f004:**
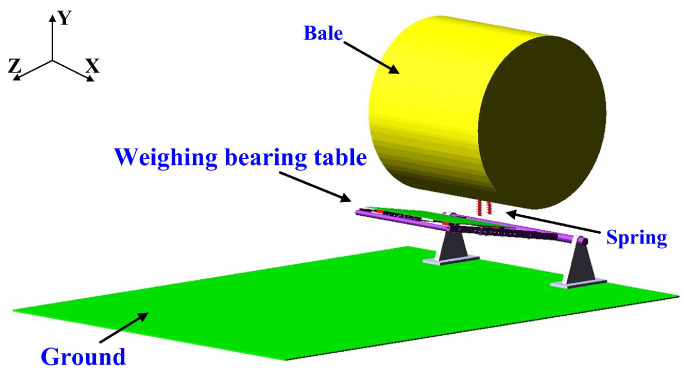
Straw bale dynamic weighing table model.

**Figure 5 sensors-23-01778-f005:**
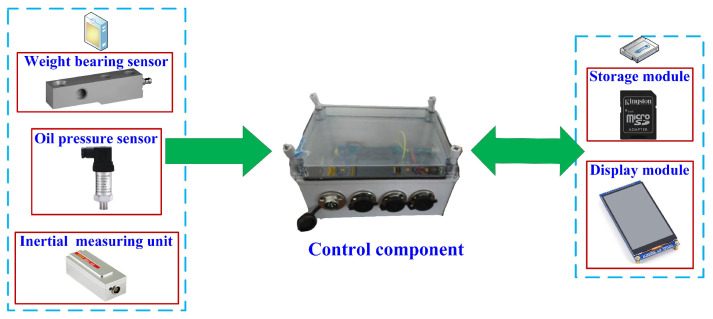
The overall structure of the monitoring system.

**Figure 6 sensors-23-01778-f006:**
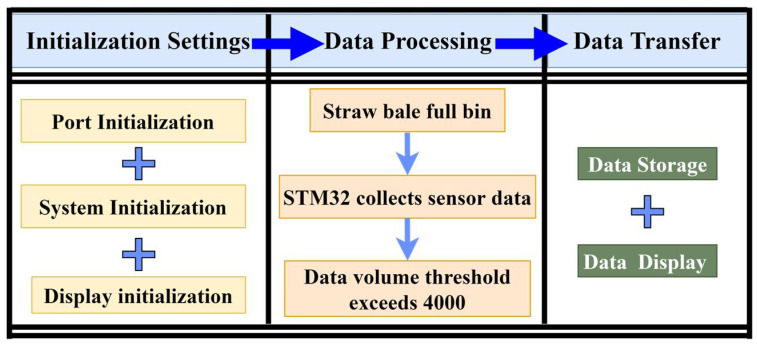
Bale density monitoring system flowchart.

**Figure 7 sensors-23-01778-f007:**
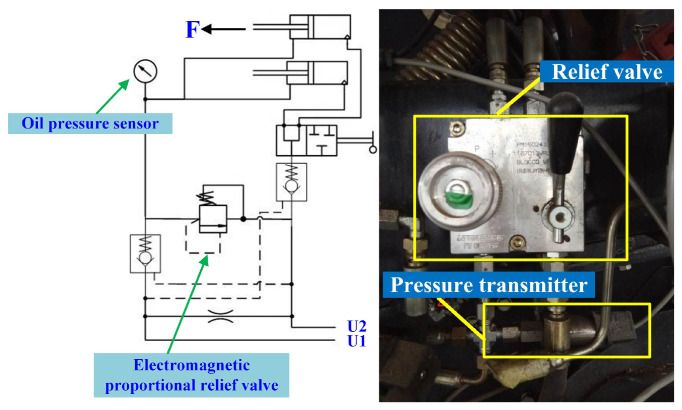
Backpack hydraulic system and pressure transmitter installation diagram.

**Figure 8 sensors-23-01778-f008:**
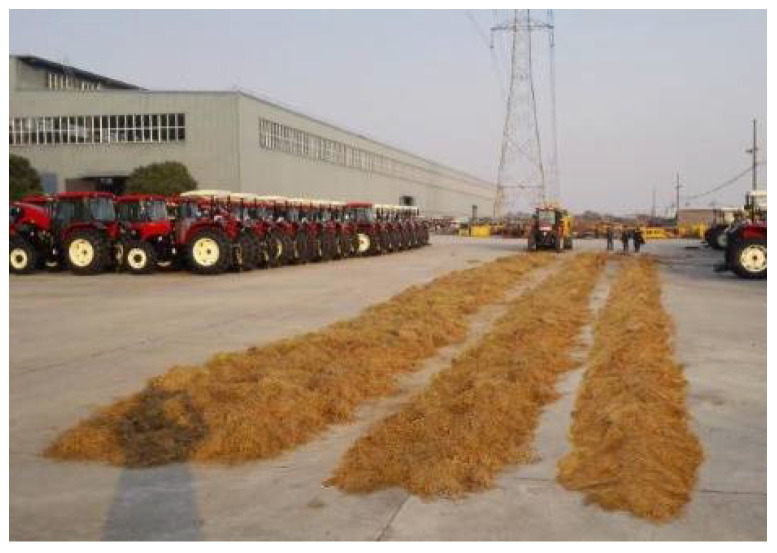
Straw laying before the experiment.

**Figure 9 sensors-23-01778-f009:**
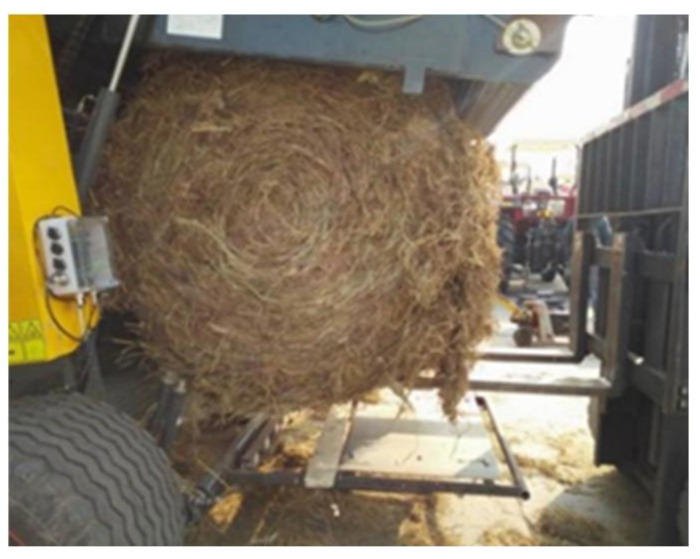
Repeatability tests of straw bale weighing.

**Figure 10 sensors-23-01778-f010:**
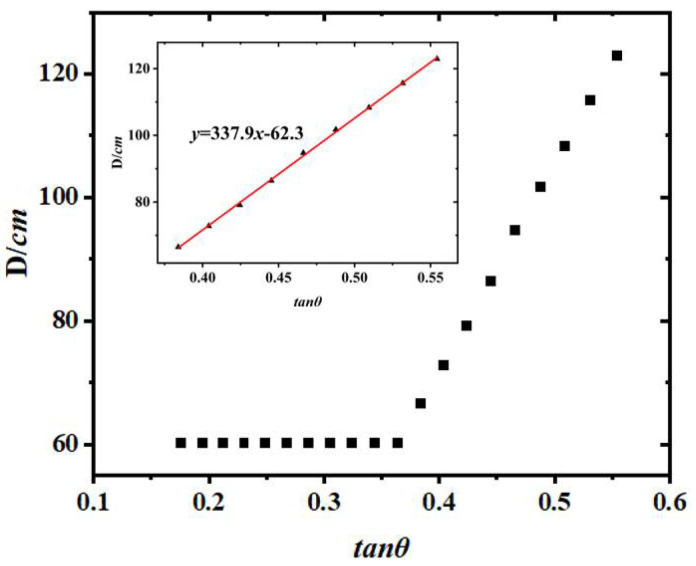
D-tan *θ* image.

**Figure 11 sensors-23-01778-f011:**
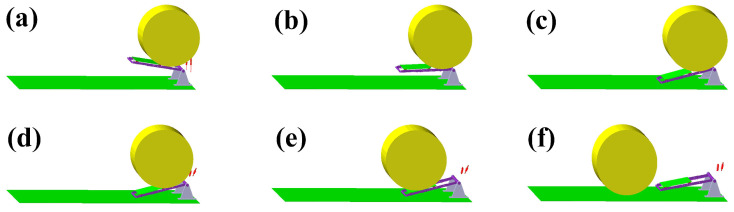
Straw bale unloading process dynamics simulation results. (**a**) Bale in contact with the table; (**b**) table swinging around the spindle; (**c**) table in contact with the ground; (**d**) bale rolling along the table mount; (**e**) bale rolling along the data collection table surface; (**f**) table swinging back.

**Figure 12 sensors-23-01778-f012:**
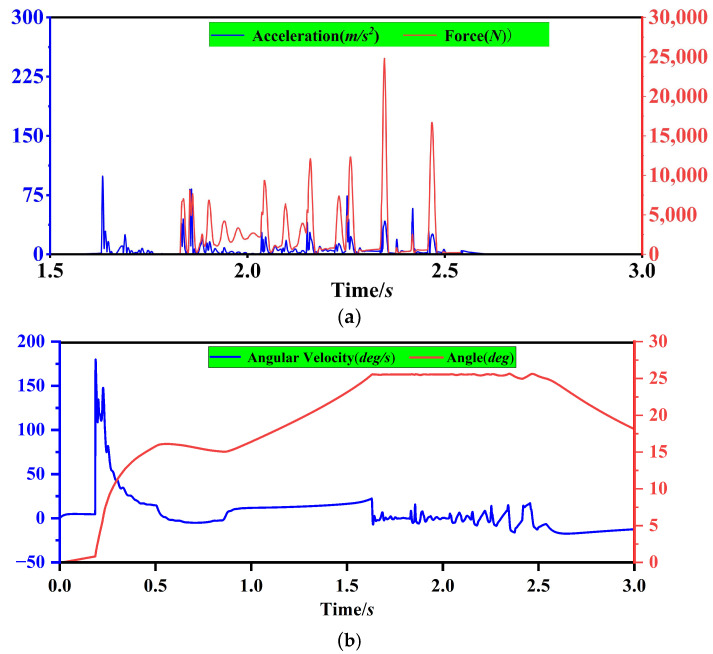
Pressure, acceleration and attitude angle of the loading table in the simulation of the dynamics of the bale unloading process. (**a**) Pressure and acceleration variation curves of the load-bearing table data acquisition table. (**b**) Angular velocity and deflection angle θ of the carrier table around the X-axis during unloading of the bale.

**Figure 13 sensors-23-01778-f013:**
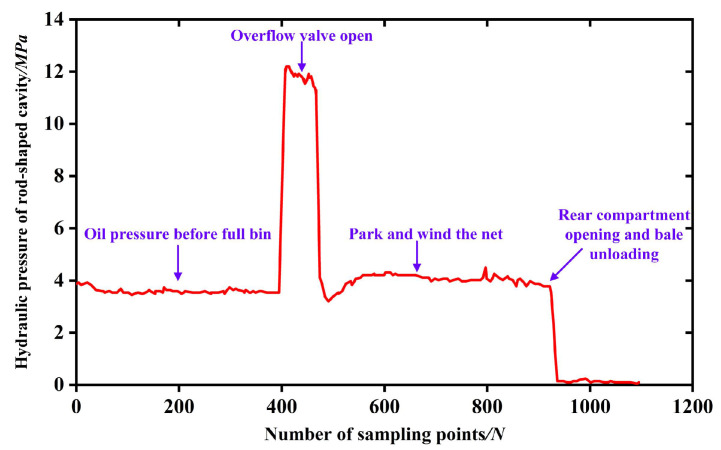
Variation in oil pressure in the rod chamber of the round baler backpack cylinder.

**Figure 14 sensors-23-01778-f014:**
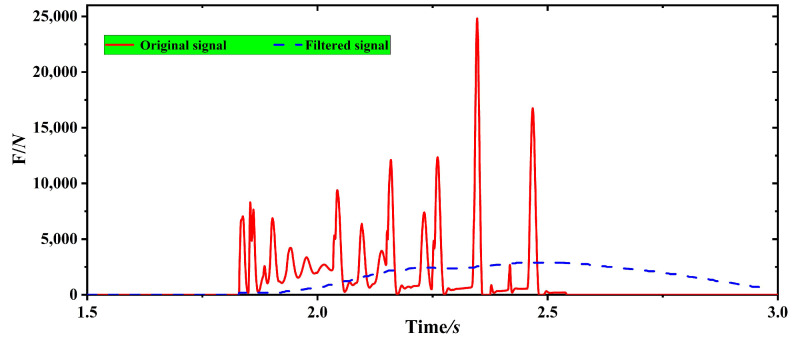
Before and after filtering of the pressure on the data acquisition table in the simulation of the dynamics of the unloading process.

**Figure 15 sensors-23-01778-f015:**
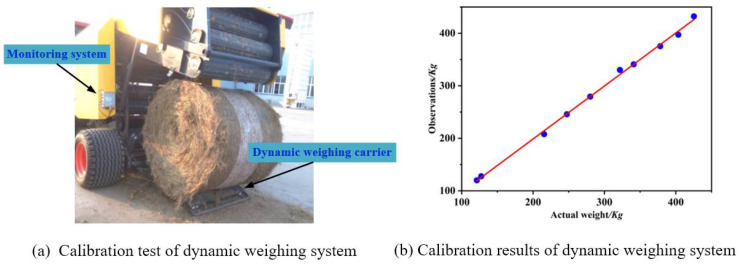
Dynamic weighing system calibration.

**Figure 16 sensors-23-01778-f016:**
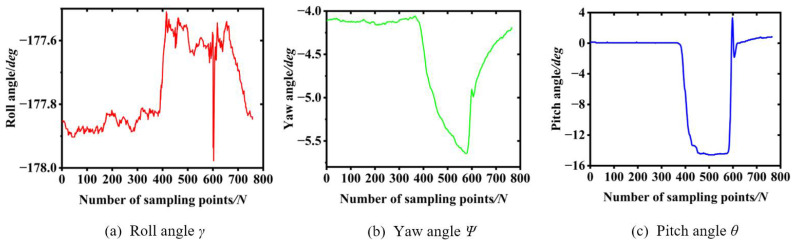
Attitude system data.

**Figure 17 sensors-23-01778-f017:**
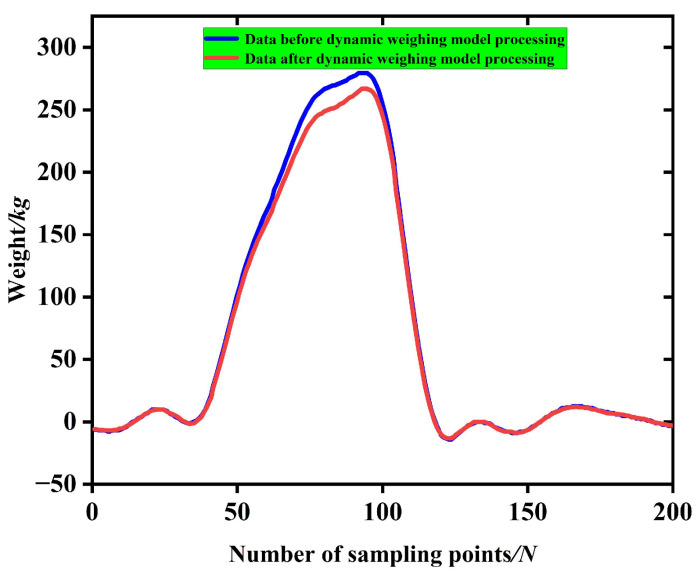
Dynamic weighing models before and after use.

**Figure 18 sensors-23-01778-f018:**
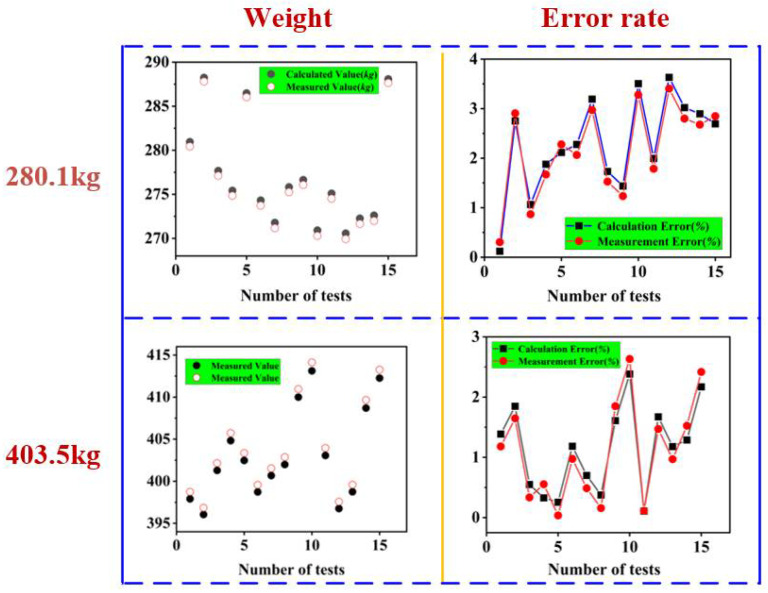
Straw bale repeatability test monitoring results and error values.

**Table 1 sensors-23-01778-t001:** Straw bale material parameters.

Straw Bale Density (kg/m^3^)	Poisson′s Ratio	Modulus of Elasticity (Mpa)
188.446	0.35	0.049

## Data Availability

All of the data generated or analyzed during this study are included in this published article.

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
