# Peer review of "Design and Experiments of a Real-Time Bale Density Monitoring System Based on Dynamic Weighing"

_sensors, 2023, doi:10.3390/s23041778_

Round 1

Reviewer 1 Report

This paper proposes a new method of bale density monitoring, which realizes the real-time monitoring of straw bale density.It not only completes the design of the bale density monitoring system, but also the simulation and experiments on the dynamics of the weighing process, which is of great practical value.However, there are some improvements that need to be made before this paper can be accepted. These include:

1. The authors should consider changing the title of the paper from Monitoring Method and Experiment of Bale Density of Round Baler Based on Dynamic Weighing to Design and experiments of a real-time bale density monitoring system based on dynamic weighing, or any other.

2.In the abstract, the sentences in lines 11 to 18 need to be simplified to make the manuscript clearer. Please revise it.

3.The sentences of lines 120 to 124 is confusing, it is suggested to written as two lines.

4. Figure 15 is not clear enough, using high solution picture instead of it, and mark each key component.

5. The conclusions are more general in nature. Novelty and importance should be highlighted.

Author Response

Dear reviewer, the specific revision comments are in the uploaded document and manuscript, please check, thank you.

Reviewer 2 Report

1. The accuracy of the measurement can meet the requirements for 24 real-time density monitoring of the volumetric fixed round bale machine and provide a basis for 25 real-time regulation of bale density.It is not precise

2. What is the significance of using  round balers rather than squared one?

3. It is not presented in detail about how STM32 processor collects teh sensor data

4. Fig12(a)  and Fig 13 are not clear

5. When the bale is full and the back-354 pack door is squeezed, the oil pressure in the rod chamber increases to 12 MPa, the bale 355 full travel switch was triggered, the relief valve connected to the rod chamber unloads 356 and the oil pressure in the rod chamber plummets. Any justification?

6. All graphs need to improve. It is not in readable form

Author Response

(The authors gave the same response as above.)

Reviewer 3 Report

This study aimed to design a dynamic weighing system for round bale machines  based on pressure detection. The work is clear and the methodology is well structured, considering that it can contribute to the state of the art of the subject.

However, I believe that some small changes should be made to improve the quality of the publication:

1.- In introduction section, it is suggested to include this work and indicate advantages and disadvantages with respect to the development in it.

https://doi.org/10.1016/j.compag.2021.106044

2.- Improve the quality of all the figures

3.- Figures from 13 to 18 increase the size of the figures and specifically the font size, it is not possible to read the content of the labels.

4.- Include a discussion of a probable future of these systems, example; considering new technologies such as piezoelectric sensors that are used in dynamic weighing on roads for example.

Author Response

(The authors gave the same response as above.)
